# An Alternative Platform for Protein Expression Using an Innate Whole Expression Module from Metagenomic DNA

**DOI:** 10.3390/microorganisms7010009

**Published:** 2019-01-08

**Authors:** Dae-Eun Cheong, So-Youn Park, Ho-Dong Lim, Geun-Joong Kim

**Affiliations:** Department of Biological Sciences and Research Center of Ecomimetics, College of Natural Sciences, Chonnam National University, Yongbong-ro, Buk-gu, Gwangju 61186, Korea; decheong01@gmail.com (D.-E.C.); soyoun.park@mail.mcgill.ca (S.-Y.P.); scott.lim1@gmail.com (H.-D.L.)

**Keywords:** protein expression, innate module, evolutionary optimization, expression vector, *Escherichia coli*

## Abstract

Many integrated gene clusters beyond a single genetic element are commonly trapped as the result of promoter traps in (meta)genomic DNA libraries. Generally, a single element, which is mainly the promoter, is deduced from the resulting gene clusters and employed to construct a new expression vector. However, expression patterns of target proteins under the incorporated promoter are often inconsistent with those shown in clones harboring plasmids with gene clusters. These results suggest that the integrated set of gene clusters with diverse cis- and trans-acting elements is evolutionarily tuned as a complete set for gene expression, and is an expression module with all the components for the expression of a nested open reading frame (ORF). This possibility is further supported by truncation and/or serial deletion analysis of this module in which the expression of the nested ORF is highly fluctuated or reduced frequently, despite being supported by plentiful cis-acting elements in the spanning regions around the ORF such as the promoter, ribosome binding site (RBS), terminator, and 3′-/5′-UTRs for gene expression. Here, we examined whether an innate module with a naturally overexpressed gene could be considered as a scaffold for an expression system. For a proof-of-principle study, we mined a complete expression module with an innately overexpressed ORF in *E. coli* from a metagenomics DNA library, and incorporated it into a vector that had no regulatory element for expressing the insert. We obtained successful expression of several inserts such as MBP, GFPuv, β-glucosidase, and esterase using this simple construct without tuning and codon optimization of the target insert.

## 1. Introduction

In many basic and application fields of biological sciences, the functional overproduction of proteins is an essential step for further studies or practical applications. Over the past few decades, many studies have attempted to obtain functionally overexpressed recombinant proteins from various hosts [1]. These efforts have mainly focused on the development of molecular biological techniques and strategies that improve the traits of host strains, incorporate a fusion partner or tag, and design novel expression vectors, thereby generating suitable tools for high-level expression of foreign proteins in heterogeneous hosts [2]. Additionally, engineering the open reading frame (ORF) with difficult-to-express properties using directed evolution or codon optimization provides an additional way to improve the protein production in vivo [3]. However, despite the accumulation of intensive knowledge on the genetics and physiology of the most popular host, *Escherichia coli*, a number of genes are not well expressed in this organism [4] and thus remain to be further resolved. Furthermore, a large number of unknown genetic materials mined from metagenome sequencing projects require alternative systems different from conventionally available systems.

Most eubacteria, including the typical expression host *E. coli*, have a typical feature of coupling gene transcription with translation, and the folding of the resulting polypeptide is occasionally coupled with these systems in temporal and spatial coordination [5]. These physicochemically linked processes are controlled by the integrated coordination of various environmental and genetic factors. From this perspective, a comprehensive interpretation of bacterial gene expression should be more complex than expected due to orchestrated regulatory mechanisms according to changes in the intracellular concentration and activity of several transcription and translation factors. During these processes, trans-acting elements distinctly interact with cis-acting elements, such as codons [6], promoters [7], operators [8], ribosome binding sites (RBSs) [9], 5′-/3′-UTRs [10], leader sequences [11], and intervening sequences [12] for gene expression. Accordingly, the above-mentioned factors should be exquisitely integrated for the successful production of foreign proteins in terms of transcriptional and translational efficiency, mRNA structure and stability, avoiding quality control, folding, and localization of the translated polypeptide, as well as lowering the physiological stress of the host. Thus, it is necessary to modulate the expression system as a complete set of factors rather than simply incorporating, substituting, and fine-tuning a single element or the restricted elements in typical expression systems. Nevertheless, almost all studies have described the development of novel expression systems by incorporating an individual component into well-known scaffolds or prototypes, for example, via modification or replacement of promoters. Although the intrinsic function and structural organization of these interacting factors in expression systems mainly attract significant attention, there are no effective analysis tools available to mine a complete set of factors for gene expression in vivo.

We previously reported a promoter-trap system to screen promoters from metagenomic DNA using a dual reporter that faced the opposite direction without any promoter in the vector pBGRI [13]. Using this system, we can screen DNA fragments including the predicted promoter regions, as well as isolate a number of large DNA fragments that reveal several ORFs and promoters, or promoter-like sequences. Among these, a large fragment was identified to contain a hyper-expressed innate ORF. In this study, we suggest a set of expression modules for heterologous expression of the recombinant protein and further attempt to construct a novel expression system by simple substitution of the target gene with the innately overexpressed ORF in this module (Figure 1). We then evaluated the capability of this module as an expression system by analyzing the expression level of foreign proteins.

## 2. Materials and Methods

### 2.1. Bacterial Strain and Plasmid DNA

*E. coli* XL1-Blue (Stratagene, La Jolla, CA, USA) was used as the host for constructing the metagenomic DNA library and expressing foreign proteins. A previously constructed trap vector pBGRI, employed as a cloning vector for the construction of a soil metagenomic DNA library [13], was used for analyzing the expression patterns of deletion fragments of trapped DNA including expression modules (Figure 1). To amplify the corresponding DNA fragment containing the innately overexpressed ORF, the recombinant plasmid isolated from the screened clone was used as the template. A set of primers described in Table 1 and high fidelity Taq polymerase, Phusion (New England laboratory, Hitchin, UK) were used for PCR. The resulting DNA was purified by using a DNA clean-up system (Promega, Madison, WI, USA). pTrc99a (Pharmacia Biotech, Uppsala, Sweden) was used as the backbone plasmid to construct the pTB vector without any factors for gene expression. pQE30-1767 [14], pTrc99a-SmGlu [15], pMal-c2x Phusion (New England laboratory, Hitchin, UK), and pGFPuv (Clontech, Mountain view, CA, USA) were used as templates for PCR amplification of esterase 1767, β-glucosidase, maltose binding protein, and green fluorescence protein, respectively.

### 2.2. Deletion and Sequence Analysis of an Expression Module

The insert of the original clone isolated from the metagenome library using a promoter-trap system based on fluorescence emission was serially deleted using appropriate restriction enzymes based on a physical map, and further deleted by serial deletion using ExoIII nuclease (Promega, Madison, WI, USA) according to the general protocol provided by the manufacturer. The resulting DNA fragments were ligated into pTB derived by PCR using a set of pTB primers (Table 1) from the vector pTrc99a. The vector pTB used in this work consisted of an origin of replication, antibiotic selection marker, and transcriptional terminator without a promoter, lacI^q^ gene, and the multicloning site of pTrc99a. During these steps, the protein expression capability of the expression module was monitored by the fluorescence emission of the innate ORF, mBFP, using a spectrofluorometer (Infiniti M200, Tecan, Männedorf, Switzerland) under excitation wavelength of 365 nm. These results were also confirmed by expression analyses using SDS-PAGE (12%) after *E. coli* transformants harboring the resulting plasmid were cultured in LB medium (50 µg/mL ampicillin) at 37 °C for 12–14 h under constant shaking at 200 rpm. Finally, the resulting trimmed fragment (BMS3) with a size of about 3 kb was selected as a candidate expression module for recombinant protein expression.

DNA sequencing was carried out by primer walking using pBGRI binding primers [13] for analyzing the structural organization of the resulting trimmed expression module. The nucleotide sequence of BMS3, deposited in GenBank database (HM352833.1), was analyzed as query using the default options of BLAST X/N at the National Center for Biotechnology Information (NCBI). The putative promoter regions were predicted using the NNPP (http://www.fruitfly.org/seq_tools/promoter.html) and Prom-Find (http://nucleix.mbu.iisc.ernet.in/prompredict/prompredict.html) programs. The regulatory protein binding sites for transcription were predicted by a sequence searching program using the basic matrix of Wconsensus (http://ural.wustl.edu/consensus/cgi-bin/Server/Interface/basic_wconsensus.cgi). A consensus matrix was prepared using the RegulonDB database with information on transcription regulation sequences (http://regulondb.ccg.unam.mx/) [16].

### 2.3. Construction of Expression Vectors and Analysis of Vector Stability

The incorporation of specific restriction enzyme sites into the expression module was performed by PCR using a set of primers (Table 1). The resulting expression module without the innate ORF (mbfp gene) was subcloned into the pTB vector yielding a new expression vector pBEM3. As a control, the promoter region of mBFP was amplified by PCR and incorporated into the pTB vector to prepare the pTB2 vector.

To construct expression vectors containing a part of the N-terminal sequence (tentatively named as the leader sequence) of the hyper-expressed protein (mBFP) in the expression module, 21 arbitrarily selected nucleotide sequences were incorporated into pBEM3 by PCR, followed by construction of the pBEM4 and pBEM5 expression vectors. The plasmid pBEM5 was inserted with one nucleotide ‘G’ into the translation start site (ATG) of the leader sequence to avoid translational fusion of this sequence with the target protein.

### 2.4. Cloning and Expression of Foreign Genes in Novel Expression Vectors

To test the feasibility of novel expression systems, four genes encoding MBP, GFPuv, esterase 1767 [14], and SmGlu [15] were amplified by PCR using appropriate primer sets from pMal-c2x, pGFPuv, pQE30-1767, and pTrc99a-SmGlu, respectively (Table 1). The PCR-amplified genes were treated with appropriate restriction enzymes and then ligated with the corresponding vector digested with the same restriction enzymes. All inserts were also subcloned into the pTrc99a, pAL-c2X (derived vector by deletion of the gene encoding MBP from pMAL-c2X), and pTB2 vectors for use as control groups.

To analyze the expression pattern of foreign proteins in recombinant cells with pBEM series vectors, recombinant cells were cultured in LB medium (50 µg/mL ampicillin) at 37 °C under constant shaking at 200 rpm. After 1% of the culture broth was reseeded into 15 mL of the same medium, the cells were further grown at 37 °C for 10–12 h without any inducer. As control groups, recombinant cells harboring the plasmids, described above, with the same inserts were also reseeded into 15 mL of the same medium and grown to an OD600 of ~0.5, and then induced with 1 mM IPTG for 90 min. The cultured cells were harvested by centrifugation at 13,000 rpm for 2 min and resuspended in 20 mM Tris-HCl buffer (pH 7.5). After washing twice with the same buffer, the resulting cells were disrupted using a sonicator according to the typical procedure. Cell debris was removed by centrifugation at 13,000 rpm for 30 min at 4 °C. The resulting supernatant was loaded on a 10% SDS-PAGE and 8% Native-PAGE gel. Zymogram-based activity staining was performed with α-naphthyl acetate (45 µg/mL) and fast blue RR (45 µg/mL) for esterase [14] and MUG for glucosidase [15].

### 2.5. Prediction of Translation Initiation Rate of Foreign ORFs in the pBEM Series Vector

To validate the effects of the leader sequence on the translation of foreign genes, we predicted the translation initiation efficiency using the ribosome binding site calculator on the Salis laboratory website (https://salislab.net/software/) [17]. To calculate the translation initiation rate, *E. coli* MG1655 was chosen as an organism parameter due to the sequence (ACCTCCTTA) at the 3′ end of 16 rRNA, which complements the typical RBS sequence (AGGA). The degenerate RBS sequence parameter used for pBEM3, pBEM4, and pBEM5 was GAGGTTGACCCAT, GAGGTTGACC, and GAGGTTGACCAGTGCAGAATCTGAACGGCAAACAT, respectively. The degenerate RBS was longer than the typical RBS (6–7 bp) that appeared in commercial expression vectors because of the transcriptional fused leader sequence. Lastly, the pre- and protein coding sequence appeared 20 bp before the RBS (GAGG) of BEM3 DNA fragments and as 50 bp encoding the foreign ORFs with a start codon, respectively.

## 3. Results and Discussion

### 3.1. Selection of a Clone Having an Innate Module with a Hyper-Expressed ORF from a Metagenomic Library

In a previous study, we designed a DNA fragment-trap vector system equipped with a trapped element (promoter)-driven reporter expression logic and screened many clones that induced green or red fluorescence depending on the trapped DNA containing promoter and/or promoter-like signals [13]. Interestingly, during sequence analyses using web-based prediction procedures, over 37% of the trapped DNA from 402 positive clones possessed elemental DNA corresponding to promoter or promotor-like signals, along with additional elements deducing several open reading frames. Thus, the corresponding size of trapped DNAs frequently exceeded more than 3–6 kb. Occasionally, over 10 kb of trapped DNA fragments were also found in clones with bright fluorescence signal resulting from DNA-driven reporter expression. These relatively larger sizes of inserts screened using DNA trap vector systems were not unusual because a single reporter-equipped trap system had also easily trapped a catabolic operon from the metagenome-based library using the SIGEX procedure [18].

Further experimental analysis of recombinant plasmids with larger inserts showed that the trapped DNAs could distinctly induce reporter expression using their own promoter or promoter-like signal sequences along with an innately overexpressed ORF in surrogate hosts without any inducer under normal conditions. As is generally known, almost all larger sized DNAs have a typical feature for structural organization of catabolic or anabolic operons. However, several DNAs with the driven ability of reporter expression and a hyper-expressed innate ORF showed no typical organization of the operon. We, therefore, assumed that these DNA fragments had a complete set of cis- and trans-acting elements for gene expression whether they were inducible or constitutive. This means that these complete sets of cis- and trans-acting elements (tentatively termed as “expression module”) could be possibly used as an expression scaffold in the screening host *E. coli*. We thus attempted further to construct the expression vector using this module. From the finally screened trapped DNAs, we arbitrary selected the clone, pBGRI-BMS, that innately had an overexpressed protein (about 25 kDa). The overexpressed protein was already identified as an NADPH-binding protein with enhanced blue fluorescence and was thus named as mBFP [19]. As previously reported, the size of the insert from pBGRI-BMS was determined to be about 10 kb, and the protein mBFP was overexpressed in the soluble fraction (~35% of total cell proteins). As a control, we primarily constructed an expression vector only using the putative promoter of mBFP. However, this conventional approach showed quite low expression of the recombinant protein, including mBFP, thereby supporting our assumption that a pre-existing set of cis- and trans-acting factors beyond the predicted promoters in the BMS fragment are required for hyper-level expression of the nested protein mBFP.

To minimize this DNA as an expression module for hyper-expression of nested mBFP, we serially deleted both ends of the 5′- and 3′-regions using exonuclease III, and the resulting constructs were retransformed into the host *E. coli* XL1-Blue for analysis of mBFP expression patterns (Figure 1). Consequently, we selected a clone harboring the recombinant plasmid with an insert size of approximately 3.2 kb as a plausible candidate expression module (BMS3) for functional overexpression of the nested mBFP and further attempted to employ this module as an expression system for a surrogate host after sequence analysis by DNA sequencing.

### 3.2. Sequence Analyses of the Minimized BMS3 DNA for Further Use as an Expression Module

To analyze the structural organization of the minimized module for expression of the nested mBFP, the complete sequence of the DNA fragment (3.2 kb) was defined and analyzed using programs available at the NCBI website. The BMS3 DNA consisted of three ORFs that were predicted as a diguanylate cyclase (identity, 73%), TetR family transcriptional regulator (identity, 79%), and short-chain dehydrogenase (identity, 67%) based on BLAST search results (Figure 2A). Among them, two sequences, other than diguanylate cyclase, were previously reported as a putative protein and a short-chain dehydrogenase mBFP [19].

To further find the control region for hyper-expression of the nested ORF encoding mBFP, we primarily predicted the promoter region using online software (NNPP and Prom-Find) and found several plausible regions around the three ORFs, especially near the region of the gene mbfp. Prediction results showed that eight promoters existed on the BMS3 DNA although their real functions were not examined (Figure 2A). Furthermore, the binding sites for transcriptionally regulatory proteins were also putatively predicted to show a recognizable sequence by transcriptional regulators CRP, FNR, and Lrp (Figure 2A). These transcriptional regulatory proteins are well known to interact with several sigma factors, such as σ70, σ54, σ38, σ32, and σ24 [20].

Some previous experimental and bioinformatics analyses have shown that a number of *E. coli* promoters including their spanning regions are recognized through coordinated interaction of multiple transcription factors, such as the interaction of regulatory proteins with the holoenzyme of RNA polymerase, as in the case of eukaryotes [21]. In line with this, the multifactor promoters are also known to be recognized by more than one of seven sigma factors. Additionally, the multi-target transcription factors participate in the regulation of transcription ranging up to hundreds of promoters for genes and operons as global regulators; thus, these transcription factors and promoters could be assembled in hierarchical networks for transcription regulation [21]. However, in silico prediction of regulatory signals for transcription in bacterial DNA still remains a difficult problem in bioinformatics because of the lack of algorithms capable of making reliable predictions, recognition, and association between sequences and transcription factors. These reports indicate that the underlined mechanism of gene expression in the bacterial system is more complex than expected, and thus support a possibility that the structural organization tuned in nature during evolution, termed as an expression module, is employable as an expression system for recombinant proteins, although functionally unknown and/or physically remote cis- and trans-acting factors are frequently found in the putative module. Based on these possibilities, we assumed that hyper-expression of the nested mBFP in BMS3 DNA was regulated by the coordinated control of several cis- and trans-acting factors located around its encoding region. Therefore, we attempted to use the whole DNA of BMS3 as an expression module for a protein expression.

### 3.3. Construction and Stability Analysis of the Expression Vector Using the BMS3 Module

To this end, we amplified BMS3 DNA except for the nested mbfp by PCR using two sets of specific primers (Table 1). The amplified DNA with specifically designed restriction sites, instead of mbfp, was subcloned into the vector pTB, thus yielding the pBEM3 expression vector (Figure 2B). During the construction procedure, a transcription termination signal was located downstream of a promoter to provide stability of the vector through avoiding physiological burden and thus preventing growth retardation of the recombinant cell caused by ambiguous transcripts produced under a promoter.

To confirm the effect of pBEM3 on cell stress, we analyzed the growth rate and observed microscopic shapes of recombinant cells harboring pBEM3, pTB2, and pTB. During monitoring of the cell density (OD600) every hour, the expression vector pBEM3 did not influence cell growth compared to the cells harboring pTB and pTB2 vectors as controls (Figure 3). Additionally, microscopic observation and SDS-PAGE analysis of cultured cells also revealed no distinct differences between the cells harboring pBEM3 and the controls (data not shown). These results indicated that the pBEM3 vector containing an expression module with several cis- and trans-acting elements had no apparent negative effect on cell growth and physiology. Moreover, the plasmid copy number of pBEM3 vector was not significantly different from that of control vectors when analyzed on agarose gel after purification from cultured cells. These results suggested that the pBEM3 vector could be used as an expression system.

### 3.4. Analysis of Recombinant Protein Expression Using the pBEM3 Vector in E. coli

To achieve foreign protein expression under control of the expression module BEM3, we constructed recombinant plasmids in which several genes encoding MBP, GFPuv, SmGlu, and esterase 1767 were incorporated into the cloning site of pBEM3. Each gene in the resulting construct was located at the same site originally occupied by the innate ORF encoding mBFP. The proteins employed here showed distinctly different expression patterns from previous reports [14,22]. All cases of foreign proteins were functionally expressed in recombinant cells, detected as the same corresponding bands from pTB2 and commercial vectors (pTrc99a and pMAL-c2X) as controls (Figure 4A). However, the expression levels of recombinant proteins from pBEM3 were relatively lower than those from commercial vectors chosen as results of optimization procedures using various commercially available vector systems, such as pET, pMAL, and pTrc99a series vectors. Intriguingly, the protein expression level was distinctly higher than that of the control vector pTB2 with only the promoter of the nested mBFP (Figure 4A). These results partly provided evidence that only the promoter region could not be enough, and that a complete module was required to overexpress the cloned gene as expected.

Although the foreign protein expression in pBEM3 vector was induced as a functional form, it did not show a comparable level to the innate mBFP in the pBEM3 vector [19]. As is generally well known, the expression level of foreign proteins in a heterologous host is dramatically influenced by their innate properties such as translation efficiency, folding landscape, and mRNA and protein stability, showing different expression levels under the control of the same expression system. These phenomena are closely linked with the 5′-nucleotide and/or amino acid sequence (occasionally termed leader sequence) of the gene and its encoded protein [23]. Therefore, we further attempted to introduce this sequence (leader sequence of mBFP) into the expression vector pBEM3.

### 3.5. Analyses of Protein Expression in Recombinant Cells with Expression Vectors pBEM4 and pBEM5

Two expression vectors, pBEM4 and pBEM5, were constructed by incorporation of the leader sequence from the gene encoding mBFP into the expression vector pBEM3. The incorporated leader sequence is ATGCAGAATCTGAACGGCAAA and its decoded amino acid sequence comprises 7 amino acids (MQNLNGK). As shown in Figure 2B, this sequence incorporated in pBEM4 was translationally coupled and thus expressed as a fusion tag. However, this sequence in pBEM5 was transcriptionally fused and did not alter the N-terminal amino acid sequences of the expressed protein. This relatively short sequence as a tag for changing the translation or transcription was arbitrary chosen based on our previous reports describing the effect of N-terminally elongated sequences or codon variants on the target protein expression level in vivo [14,24].

Considering the expression profiles of foreign proteins from pBEM3, the relatively low-level expressed genes encoding esterase 1767, GFPuv, and SmGlu were subcloned into two expression vectors, pBEM4 and pBEM5, respectively. The resulting expression level of GFPuv was dramatically increased in the clone harboring pBEM4 and the expression levels of SmGlu were also distinctly increased in clones harboring pBEM4 and pBEM5 (Figure 4B). Unexpectedly, the band corresponding to esterase 1767 was not obviously detected in the clone harboring each of the two expression vectors. Cases of successful overexpression of recombinant protein from pBEM4 and pBEM5 were probably due to the positive effects of the fused leader sequence on mRNA stability, interaction with the folding chaperon, and translation rate. To indirectly ascertain effects of leader sequences on translation efficiency, we calculated the translational initiation rate using the RBS calculator [17]. In the pBEM4 vector, both results from SDS-PAGE analysis and theoretically calculated values were well correlated. However, in pBEM5, bioinformatically calculated values was apparently different from the SDS-PAGE results (Table 2). These results might be attributed to the atypically long RBS generated by transcriptional fusion, and thus imply that the existing calculation tool based on accumulated knowledge is still insufficient for the comprehensive analysis of gene expression. Currently, we are systematically analyzing the expression level and patterns of the recombinant protein from pBEM series vectors to introduce these systems as an alternative for protein expression. Although further studies explaining the plausible mechanism and application of these systems to various target genes are needed, some positive results suggest that the recombinant protein could be expressed by the coordinated action of constitutive components in the complete expression module.

As described above, we did not engineer the BMS3 DNA for use as an expression module for recombinant proteins, and thus employed the innately organized structure in nature. To the best of our knowledge, even though further tuning and/or trimming remains, this is the first study in which the complete expression module from metagenomic DNA was used as an alternative expression system. Furthermore, this study suggested that successful expression is the combinatorial effect of promoters as well as the sequences spanning this region including various cis- and trans-acting elements. This effort would provide a basis to create a catalogue of alternative gene expression systems and supply the extended part for recombinant protein production.

## Figures and Tables

**Figure 1 microorganisms-07-00009-f001:**
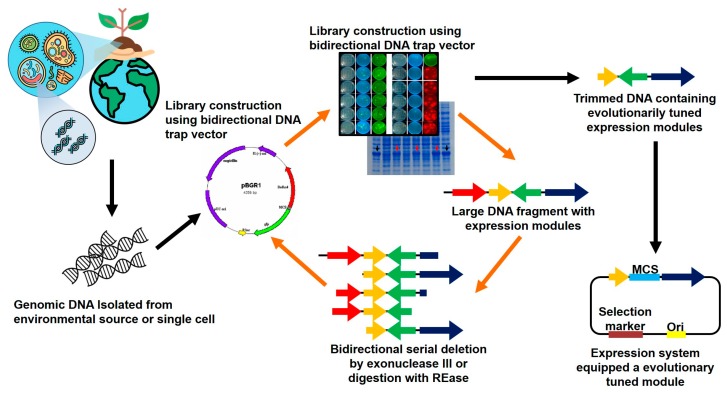
Schematic representation of the screening and trimming of expression modules from the metagenomics DNA library. During the procedure, a promoter or promoter-like signal-trap vector equipped with a bi-directional reporter system was employed. Trimming of screened DNA fragments was conducted by specific deletion or serial deletion using restriction enzymes or exonuclease III according to the general procedure. Functional analyses of reporter expression by trimmed DNA containing the promoter or promoter-like signal was monitored by UV excitation and further confirmed by expression level using SDS-PAGE.

**Figure 2 microorganisms-07-00009-f002:**
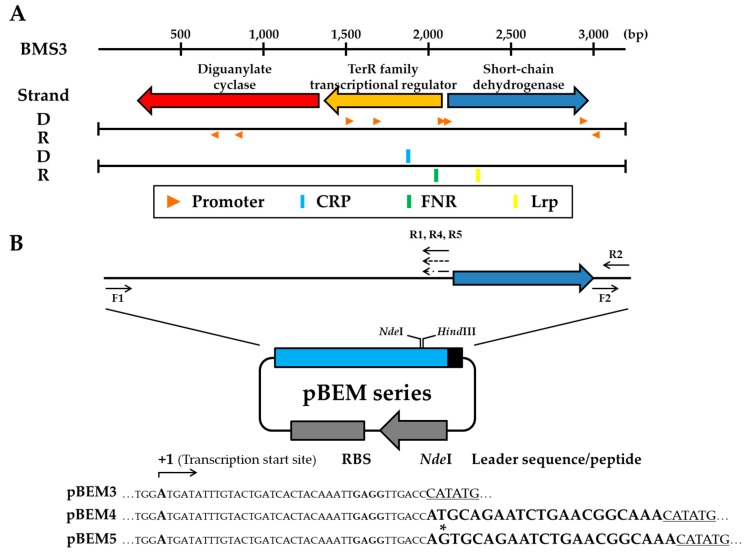
Analyses of structural organization of the BMS3 DNA sequence and construction of pBEM series expression vectors. (**A**) Deduced ORFs in BMS3 DNA. The innate overexpressed ORF in BMS3 DNA was identified as a gene encoding a short-chain dehydrogenase mBFP. The promoters and binding regions for regulatory proteins predicted by using methods described in Materials and Methods were also shown in the BMS3 DNA. (**B**) A physical map of the new expression system (pBEM series vector). pBEM3 vector was constructed using BMS3 DNA without the *mbfp* (short-chain dehydrogenase) gene. pBEM4 and pBEM5 included an arbitrary leader sequence or peptide of the 5′ sequence of the gene encoding mBFP. All vectors have the same transcription start site and RBS (GAGG). However, due to the changing of ATG to AGTG in the leader sequence of pBEM5, these two expression vectors have different translation initiation regions. The blue and black box indicates BMS3 DNA without the *mbfp* and transcription terminator, respectively. The gray box indicates the antibiotic selection marker with its own promoter (arrowed).

**Figure 3 microorganisms-07-00009-f003:**
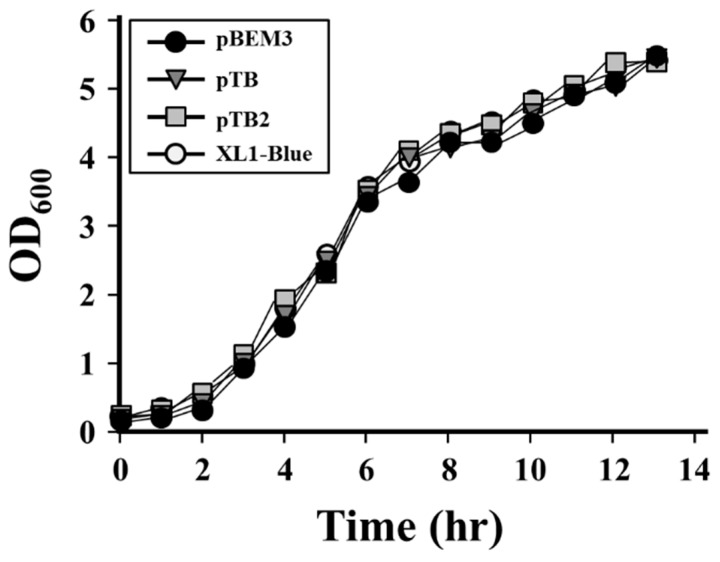
Comparison of growth curve of recombinant cells harboring the new expression vectors. The recombinant cell harboring pTB2 expression vector was used as a control.

**Figure 4 microorganisms-07-00009-f004:**
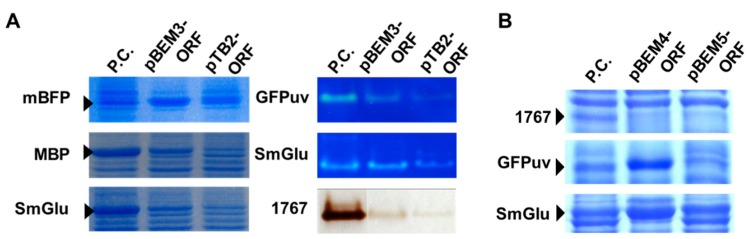
Expression analysis of foreign proteins in pBEM series vectors. (**A**) To validate the functionality of the expression module, we analyzed the expression levels of foreign proteins by SDS-PAGE, and conducted zymogram analyses using overlaid gels containing appropriate substrates according to the general procedure. P.C.: positive controls. pBGRI-BMS, pMal-c2x, pTrc99a-SmGlu, pQE30-1767, and pGFPuv were used as positive controls for mBFP, MBP, SmGlu, 1767, and GFPuv, respectively. The control panel of 1767 was run on the different gel and then artificially fused together for a clearer comparison. The expression vector pTB2 containing only the promoter regions of the *mbfp* gene was also used as a control. (**B**) Analyses of expression levels of foreign proteins in pBEM4 and 5 vectors. The leader sequence or peptide of the *mbfp* gene was incorporated into these expression vectors as described in the experimental section.

**Table 1 microorganisms-07-00009-t001:** List of primers used for amplification of several DNA sequences in this work.

Primer	Sequence (5′→3′)	REase
BMS3 DNA F1	TTTGAATTCCTCGCCGCC ^1^	*Eco*RI
BMS3 DNA R1	TTTGATATCCATATGGGTCAACCTCAAT	*Eco*RV, *Nde*I
BMS3 DNA F2	TTTAAGCTTGCGTGCAGGGC	*Hind*III
BMS3 DNA R2	TTTGAATTCTGGTGGGCTGTGAG	*Eco*RI
pTB F	ATATCTAGAGGCTGTTTTGGCGGA	*Xba*I
pTB R	TATCAGCTGCGGTGTGAAATACC	*Pvu*II
BMS promoter F	TTTGTGTTGATCGATAAGAAAATC	
BEM4 common R	ATACATATGTTTGCCGTTCAGATTCTGCAT	
BEM5 common R	ATACATATGTTTGCCGTTCAGATTCTGCAC ^2^T	*Spe*I
mBFP F	TATCATATGCAGAATCTGAACGGCAAAGTGGCTT	*Nde*I
mBFP F	TATAAGCTTTCAAGCGGCGAAGCC	*Hind*III
MBP F	CGCCATATGAAAATCGAAGAAGGTAAAC	*Nde*I
MBP R	AAACATATGTCATCCGCCAAAA	*Nde*I
GFPuv F	TTTAAGCTTAAAGGAGAAGAACTTTTCACTG	*Hind*III
GFPuv R	TTTAAGCTTTTATTTGTAGAGCTCATCC	*Hind*III
SmGlu F	CCCATATGATGATCGAAGCCAAG	*Nde*I
SmGlu R	ATAAGCTTTCATCCCGGCTTGT	*Hind*III
1767 F	ATACATATGGTGCAGATTCAGGGT	*Nde*I
1767 R	TATAAGCTTTTACAGACAACCGGC	*Hind*III

^1^ The underlined DNA sequences indicate those recognized by restriction enzymes; ^2^ The bold letter indicated one nucleotide ‘G’ which was inserted into the translation start site (ATG) of the leader sequence to avoid translational fusion with the target protein.

**Table 2 microorganisms-07-00009-t002:** Translation initiation rates of foreign proteins in each of the pBEM series vectors.

Vector	RBS Sequence	ORF	Translation Initiation Rate
pBEM3	GAGGTTGACCCAT	Mbfp	5796.89
		GFPuv	724.78
		SmGlu	7358.43
		Esterase 1767	420.45
pBEM4	GAGGTTGACC	GFPuv	5633.32
		SmGlu	2472.22
		Esterase 1767	1104.66
pBEM5	GAGGTTGACCAGTGCAGAATCTGAACGGCAAACAT	GFPuv	20.9
		SmGlu	6.66
		Esterase 1767	6.66

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
