# Peer review of "An Alternative Platform for Protein Expression Using an Innate Whole Expression Module from Metagenomic DNA"

_microorganisms, 2019, doi:10.3390/microorganisms7010009_

Reviewer 1 Report

An alternative platform for protein expression using an innate whole expression module from metagenomic DNA by Cheong et al.

This work is focused on the study of transcriptional regulation and in particular on the identification and characterization of the so called “expression modules” that in addition to a native ORF, include also cis- and trans-acting elements necessary to modulate the expression of target proteins. Thus, the main goal is to develop novel expression vectors based on naturally over-expressed proteins within these modules. Although this idea is quite intriguing, the authors with the present study are far away to reach their aim. However, this article could contribute to elucidate some aspects of the complex mechanisms governing gene expression in bacteria.

Major points

A general criticism to this work could be that promoters, regulatory sequences, translation initiation rate and putative global regulators acting on BMS3 DNA have been predicted only in silico, and for example, no primer-extension experiments to map active promoters and band shift assays (EMSA) are provided.

GFP and SmGlu proteins are clearly over-expressed from pBEM4 and pBEM5 (Fig. 4B). At least for SmGlu, the expression is apparently higher than that observed in the left panel of Fig. 4A, assuming to be under same experimental conditions. Thus, in the absence of a control sample (levels of the target proteins from any commercial expression vector) the comparison is quite difficult (see also the minor point relative to Fig. 4). This control should be provided.

Fig. 4B shows that not in all cases, the leader sequence positively affects the expression of the target protein suggesting that, right now, this alternative expression system cannot be applied to any gene (see 1767 ORF). In addition, computer data (Table 2) and SDS-PAGE are confident for pBEM4 but not pBEM5 (lines 333-335). In fact pBEM5 supports very low initiation rates regardless of the ORF cloned (Table 2) and the protein levels of both GFPuv and SmGlu are significantly reduced in pBEM5 as compared to pBEM4 (Fig. 4B). Thus, to understand the role of this leader sequence becomes crucial. Using Mfold program you can verify that the insertion of a G in the leader sequence causes the formation of a very stable stem-loop structure in the 5’UTR of pBEM5 mRNAs (see below). Reasonably, this highly structured region might affect both mRNA stability and accessibility of ribosomes to the transcript. Notably, the SD sequence is located in a single stranded region in pBEM4 while is hidden within a stem structure in PBEM5. If regulation occurs at translational level, comparable levels of mRNA between pBEM4 and pBEM5 should be obtained in Northern blotting analysis. This experiment control is easy and informative. In addition mRNA stability could be measured following RNA degradation in vivo, in a time course experiment, after treatment of cells (transformed with pBEM4 and pBEM5) with rifampicin to stop transcription. Then, RNA can be detected by Northern blot.

Minor points

To make easier the interpretation of results, the legends of Figs. 2 and 4 should be more detailed. The pBEM3 construct (Fig. 2B) consists of the entire regulatory region upstream the mBFP gene. Is it right? For example, no indications are provided for blue, black and grey boxes in Fig. 2B. What does P.C. indicate in Fig. 4AB? Maybe commercial expression vectors?

The position of the terminator inserted into pEM3 should be provided (lines 263-267).

Usually, in vivo, Escherichia coli, in silico are written in italic.

The use of a soil metagenome should be mentioned in line 89 and/or in Fig. 1.
pBEM4 pBEM5

Author Response

Reviewer 1

An alternative platform for protein expression using an innate whole expression module from metagenomic DNA by Cheong et al.

This work is focused on the study of transcriptional regulation and in particular on the identification and characterization of the so called “expression modules” that in addition to a native ORF, include also cis- and trans-acting elements necessary to modulate the expression of target proteins. Thus, the main goal is to develop novel expression vectors based on naturally over-expressed proteins within these modules. Although this idea is quite intriguing, the authors with the present study are far away to reach their aim. However, this article could contribute to elucidate some aspects of the complex mechanisms governing gene expression in bacteria.

Major points

A general criticism to this work could be that promoters, regulatory sequences, translation initiation rate and putative global regulators acting on BMS3 DNA have been predicted only in silico, and for example, no primer-extension experiments to map active promoters and band shift assays (EMSA) are provided.

Response

-          We thank reviewer indeed for the careful reading and then valuable comments on generally acceptable standards of scientific article. As the reviewer mentioned above, there are various molecular biological tools to analyze and prove the interaction of cis-acting DNA elements with trans-acting transcription factors. However, almost all of these methods are conducted by artificially designed procedure in-vitro using limited (defined) sources of plausible factors which were typically deduced from the prediction algorism and/or the results of pre-existing reports. When experimentally screened factors, such as promoter and RBS (ribosomal binding site), are usually employed to construct expression systems after these works, it has frequently observed in various reports that the expression of gene of interest was highly fluctuated and also showed unexpected patterns. Accordingly, we thought that the development of the efficient expression system was suffered from the incomplete and theoretical analyses of transcription factors on DNA fragments. In this situation, we intentionally focused on the availability of naturally evolved expression modules without further analysis for modifications. These points were fully described in our original manuscript (line 46-line 66).

 GFP and SmGlu proteins are clearly over-expressed from pBEM4 and pBEM5 (Fig. 4B). At least for SmGlu, the expression is apparently higher than that observed in the left panel of Fig. 4A, assuming to be under same experimental conditions. Thus, in the absence of a control sample (levels of the target proteins from any commercial expression vector) the comparison is quite difficult (see also the minor point relative to Fig. 4). This control should be provided.

Response

-          According to the comment of reviewer, each positive control was incorporated in the revised version of manuscript (Fig. 4).

Fig. 4B shows that not in all cases, the leader sequence positively affects the expression of the target protein suggesting that, right now, this alternative expression system cannot be applied to any gene (see 1767 ORF). In addition, computer data (Table 2) and SDS-PAGE are confident for pBEM4 but not pBEM5 (lines 333-335). In fact pBEM5 supports very low initiation rates regardless of the ORF cloned (Table 2) and the protein levels of both GFPuv and SmGlu are significantly reduced in pBEM5 as compared to pBEM4 (Fig. 4B). Thus, to understand the role of this leader sequence becomes crucial. Using Mfold program you can verify that the insertion of a G in the leader sequence causes the formation of a very stable stem-loop structure in the 5’UTR of pBEM5 mRNAs (see below). Reasonably, this highly structured region might affect both mRNA stability and accessibility of ribosomes to the transcript. Notably, the SD sequence is located in a single stranded region in pBEM4 while is hidden within a stem structure in PBEM5. If regulation occurs at translational level, comparable levels of mRNA between pBEM4 and pBEM5 should be obtained in Northern blotting analysis. This experiment control is easy and informative. In addition mRNA stability could be measured following RNA degradation in vivo, in a time course experiment, after treatment of cells (transformed with pBEM4 and pBEM5) with rifampicin to stop transcription. Then, RNA can be detected by Northern blot.

Response

-          We also basically agree with the comment of reviewer. As mentioned by reviewer, the structure of the region including 5’ untranslated and N-terminal region is one of well-known factor which affects gene expression. To optimize and redesign the expression module, of course, additional data including identification of promoter, quantification and stability analysis of the transcript would be needed. However, as you know, the comprehensive analysis of expression factors in DNA fragments could not always lead to positive outcomes of generating good expression systems. Recently, Cambray et. al. (Evaluation of 244,000 synthetic sequences reveals design principles to optimize translation in Escherichia coli. Nat Biotechnol 2018, 36, 1005-1015) reported a comprehensive analysis of these factors, including transcript structure, ramp, AT content, global and local usage bias of codon, all of which affected translation in E. coli. Consequently, they provided that transcript structure and codon usage are the major factors affecting translation. However, they demonstrated that just 53% of total variance in protein production was explainable by these major factors. In addition, they also found that unrelated sequences with similar structure profiles produced different expression level, and thus speculated that these results were attributed to a poor ability to predict nucleic acid structures and their dynamics accurately in vivo. In these context, we proposed the simple method for the construction of expression systems without time consuming and laborious analysis step. Further identification of transcription factors and quantification of transcript could help us to develop a more useful expression vector, but might go beyond the scope of this paper.

Minor points

To make easier the interpretation of results, the legends of Figs. 2 and 4 should be more detailed. The pBEM3 construct (Fig. 2B) consists of the entire regulatory region upstream the mBFP gene. Is it right? For example, no indications are provided for blue, black and grey boxes in Fig. 2B. What does P.C. indicate in Fig. 4AB? Maybe commercial expression vectors?

Response

-          We carefully led the legends of Fig. 2 and 4, and then revised according to the comment of reviewer. .

The position of the terminator inserted into pEM3 should be provided (lines 263-267).

Response

-          For a clearer understanding, the terminator position was depicted in Figure 2 of the revised version of manuscript.

Usually, in vivo, Escherichia coli, in silico are written in italic. 

Response

-          After careful reading, all related words were corrected in the revised manuscript.

The use of a soil metagenome should be mentioned in line 89 and/or in Fig. 1.

Response

-          The word “soil metagenome” was incorporated in line 89, according to the comment of reviewer, but DNA source in Figure 1 means generally accessible environmental samples.

Reviewer 2 Report

This study attempted to use native promoter and ORF arrangement to express recombinant proteins. Although there is no certain conclusion for the immediate application, since the experiments shown are very limited and the conditions still need to be optimized, the idea seems worthy to be explored and may provide readers with new insight into designing protein expression vectors.

1) It is not clear why BEM3 region is selected. Please elaborate the rational and provide some useful tips for selecting a useful region.

1) In part of 3.4, for testing foreign protein expression level in pBEM3 vector, please give more explanation on why you chose 'MBP, GFPuv, SmGlu and esterase 1767' these four genes? For example, the functions and advantages of these genes.

2) In figure 4A, please indicate the meaning of P.C. labeled in the first lane and point out which lane is the pTB2 containing only the 302 promoter regions of mbfp gene. Also, do you have the SDS-PAGE result that indicating these foreign protein expression level in commercial vectors (pTrc99a and pMAL-c2X) you mentioned in part 3.4?  

3) In part 3.5, please explain more why the insertion of the leader sequence from the gene encoding mBFP into pBEM4 vector didn't affect expressed protein because as I see, it seemly altered the N-terminal amino acid sequences of target gene.

4)  In Fig 4A,why the 1767 panel seems to be composed from different gels? If samples are not run on the same gel, should be boarder lines between samples.

5) Should the data in table be from multiple repeats and be presented as average with standard deviation? 

Author Response

This study attempted to use native promoter and ORF arrangement to express recombinant proteins. Although there is no certain conclusion for the immediate application, since the experiments shown are very limited and the conditions still need to be optimized, the idea seems worthy to be explored and may provide readers with new insight into designing protein expression vectors.

1)      It is not clear why BEM3 region is selected. Please elaborate the rational and provide some useful tips for selecting a useful region.

Response

-          As described in the section 2.2 (line 105-118) of Materials and Methods (also please see Figure 1), to select the region BEM3, we constructed DNA library by digestion of BMS (originally screened insert) with restriction enzyme and /or serial deletion with ExoIII nuclease. After the transformation into the host E. coli, we finally screened a clone that had BEM3 DNA with brightly blue fluorescence emitting from overexpressed protein mBFP. Because the innate ORF encoded blue fluorescence protein mBFP as a reporter for overexpression, we could readily screen the minimized region (BEM3) to be maintain the expression level. If innately overexpressed ORF was unsuitable to use as a reporter, the substitution with the ORF encoding readily available reporter protein such as GFP, DsRed or mCherry can be alternatively chosen for screening expression modules as described in Figure 1.

2) In part of 3.4, for testing foreign protein expression level in pBEM3 vector, please give more explanation on why you chose 'MBP, GFPuv, SmGlu and esterase 1767' these four genes? For example, the functions and advantages of these genes.

Response

-          As model proteins for our proof of concept study and validation of the resulting expression vectors, we arbitrarily used MBP, GFPuv, SmGlu and esterase 1767 due to the clear difference in expression profiles of each protein. As generally well known, maltose binding protein (MBP) is well expressed in various expression system. GFPuv is also well expressed, but frequently revealed a relatively low expression level due to concentration-dependent oligomeric states. On the contrary, we previously reported that the expression level of SmGlu and 1767 were relatively low or prone to aggregation under the control of the typical expression vectors. Two incorporated references, ref 14 and 22, in the original manuscript showed these results. These points were also described in the original manuscript (line286-290). 

3) In figure 4A, please indicate the meaning of P.C. labeled in the first lane and point out which lane is the pTB2 containing only the 302 promoter regions of mbfp gene. Also, do you have the SDS-PAGE result that indicating these foreign protein expression level in commercial vectors (pTrc99a and pMAL-c2X) you mentioned in part 3.4?  

Response

- According to the comment of reviewer, Fig. 4 and its legend were revised in a new version of manuscript.

4) In part 3.5, please explain more why the insertion of the leader sequence from the gene encoding mBFP into pBEM4 vector didn't affect expressed protein because as I see, it seemly altered the N-terminal amino acid sequences of target gene.

Response

- As pointed out by the reviewer, the incorporated leader sequence (MQNLNGK) of mBFP into the expression vector pBEM4 was translationally coupled and thus fused with the target protein when expressed. The resulting tagging protein could have different nature of the stability from the wild type protein, as those observed frequently in the target protein with His-tag, FLAG-tag or Myc tag. As generally well known, a small size of peptide tag was broadly employed as an addressing and reporting tag at the N-terminal region of the target proteins. The effect of this tag on the expression level and property of expressed protein are quietly different in each case, because additionally fused tag also affected the secondary structure of the resulting transcript.

5)  In Fig 4A, why the 1767 panel seems to be composed from different gels? If samples are not run on the same gel, should be boarder lines between samples.

Response

- As pointed out by the reviewer, the control panel of 1767 was run on the different gel and then fused together for a clearer comparison. We revised the legend of Fig. 4A to incorporate this point in the new version of manuscript.

6) Should the data in table be from multiple repeats and be presented as average with standard deviation? 

Response

- Data presented in Table 2 was theoretically predicted using RBS calculator as described in the section of Materials and Methods. RBS calculator available at the website of Salis lab provides predicted results without standard deviation.

Round  2

Reviewer 1 Report

No comments.

Reviewer 2 Report

I have no extra comment.